



# Use of multiple reference data sources to cross validate gridded snow water equivalent products over North America

Colleen Mortimer[1], Lawrence Mudryk[1], Eunsang Cho[2], Chris Derksen[1], Mike Brady[1], Carrie Vuyvich[3]

[1]Climate Research Division, Environment and Climate Change Canada, Toronto, Canada
[2] Ingram School of Engineering, Texas State University, San Marcos, TX, USA
[3] Hydrological Sciences Laboratory, NASA Goddard Space Flight Center, Greenbelt, MD, USA

*Correspondence to*: Colleen Mortimer (Colleen.Mortimer@ec.gc.ca)

**Abstract.** We use snow course and airborne gamma data available over North America to compare the validation of gridded snow water equivalent (SWE) products when evaluated with one reference dataset versus the other. We assess product
performance across both non-mountainous and mountainous regions, determining the sensitivity of relative product rankings and absolute performance measures. In non-mountainous areas, product performance is insensitive to the choice of SWE reference dataset (snow course or airborne gamma): the validation statistics (bias, unbiased root mean squared error, correlation) are consistent with one another. In mountainous areas, the choice of reference dataset has little impact on relative product ranking but a large impact on assessed error magnitudes (bias and unbiased root mean squared error).
Further analysis indicates the agreement in non-mountainous regions occurs because the reference SWE estimates themselves agree up to spatial scales of at least 50 km, comparable to the grid spacing of most available SWE products. In mountain areas, there is poor agreement between the reference datasets even at short distances (< 5 km). We determine that differences in assessed error magnitudes result primarily from the range of SWE magnitudes sampled by each method, although their respective spatiotemporal distribution and elevation differences between the reference measurements and grid
centroids also play a role. We use this understanding to produce a combined reference SWE dataset for North America, applicable for future gridded SWE product evaluations and other applications.

## 1 Introduction

Snow water equivalent (SWE) is an essential climate variable critical to determining freshwater availability in montane and northern regions (Clark et al., 2001; Barnett et al., 2005). Accurate estimates of SWE are key to the verification of seasonal
forecasts (Sospedra-Alfonso et al., 2016), skilled streamflow predictions particularly at long lead times (De Roo et al., 2003; Liu et al., 2012; Wood et al., 2016), and efficient hydropower operations (Turcotte et al., 2007; Magnusson et al., 2020). Long-term spatially complete SWE records are necessary for climate assessments (e.g., Mudryk et al., 2022), effective water management (Ralph et al., 2014), and flood prediction (Vionnet et al., 2020).



Numerous publicly available gridded SWE products exist, generated from a variety of approaches ranging from earth
observation (EO) (e.g. Luojus et al., 2021), reanalysis products (e.g. Hersbach et al., 2020), snow models of varying
complexity forced by reanalysis data (e.g. Brun et al., 2013), and data assimilation schemes (e.g. Zeng et al., 2018).
Assessment of the quality of these products faces two challenges. First, there are few independent reference datasets with
long time series and well-distributed spatial coverage across the range of snow-climate zones. Second, even where and when
reference data are available, it is challenging to apply in a meaningful way because of the spatial mismatch with the typically
coarse resolution of the gridded SWE products.

Point-based SWE measurements from snow pillows (Beaumont, 1965), snow scales (Johnson, 2004; Smith et al., 2017) and
passive gamma radiation sensors (Kodama et al., 1979; Paquet et al., 2008) provide continuous records of SWE at a specific
location. However, the considerable spatial variability of SWE means that these point-based measurements are of
questionable value when applied to larger areas (Meromy et al., 2013), and are thus not suitable for the evaluation of
relatively coarse-scale gridded data. Snow courses on the other hand, consist of multiple measurements along a transect
several hundreds of meters to kilometres in length that are averaged together to provide a single SWE value (WMO, 2018).
These measurements better sample the sub-grid scale variability than a single point measurement and so are more effective
in capturing the larger-scale average. As a result, snow course data can effectively discern subtle differences in performance
between SWE products (Mortimer et al., 2022). SWE estimates from airborne gamma surveys (which measure the
attenuation of water mass by naturally emitted gamma radiation) are averaged across 300 m wide footprints and along 15–20
km long flight lines. Like snow courses, they also effectively capture the larger scale average and are appropriate to assess
the accuracy of gridded SWE products (Cho et al., 2019; 2020).

Snow course reference measurements are unevenly distributed in space and time (Fig. 1) and, as such, may not sample the
complete range of naturally occurring SWE values. If a particular dataset is tuned to a specific environment or performs
better across a certain range in SWE, the differing spatiotemporal distributions and sampled SWE ranges of separate
reference datasets could influence the determination of product performance. For example, some EO-based products have
reasonable performance up to approximately 150 mm SWE (Pulliainen et al., 2006; Luojus et al., 2021) and so perform well
against a reference dataset composed primarily of low and moderate SWE values, but will have poorer performance when
validated using a reference dataset which samples across regions with higher SWE.



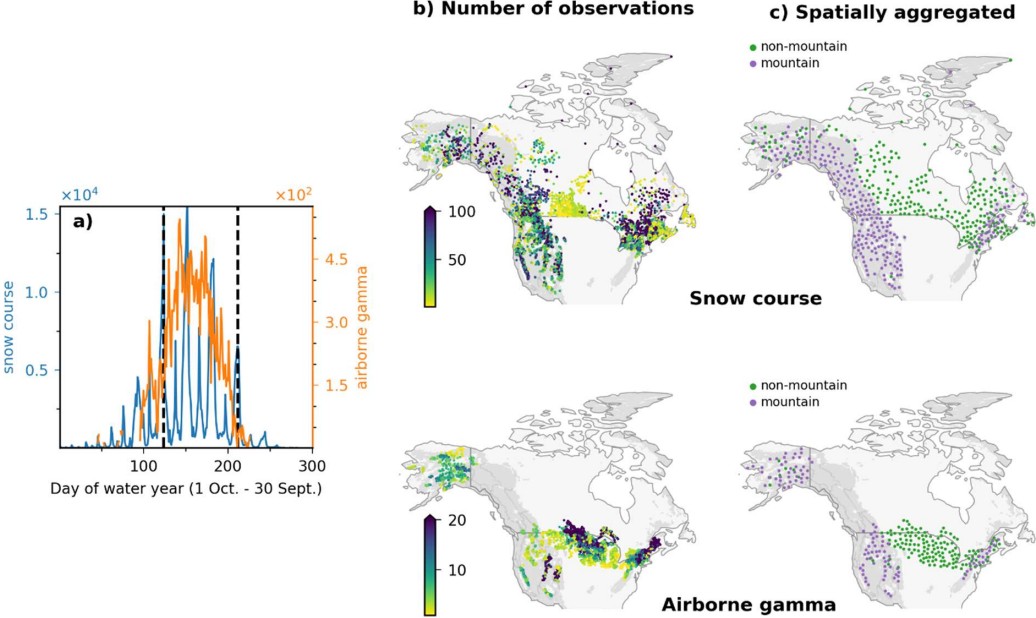


**Figure 1: Reference data distribution. (a) total number of measurements by day of water year (1 October – 30 September); vertical dotted lines delineate Feb–April period. (b) number of snow course (top) and airborne gamma (bottom) measurements during February through April 1980–2020. (c) 100 km spatially aggregated reference data separated into mountain (purple) and non-mountain (green) domains. Grey shading in b and c indicates mountain region.**

Independent assessments of gridded SWE products using either snow courses (e.g. Mortimer et al., 2020; 2022) or airborne gamma SWE (e.g. Cho et al., 2019; 2020) have been conducted, but a unified assessment of gridded SWE products using both reference datasets is lacking. Combining multiple reference datasets can improve the rigour of such assessments but interpreting and reconciling product accuracies obtained with multiple reference datasets is hindered by their differing sampling methodologies and limited uncertainty characterization, as well as their spatiotemporal distributions. Here, we
advance the evaluation of coarse resolution SWE products by using both snow courses and airborne gamma SWE estimates to evaluate gridded SWE products over North America. We investigate the agreement in reference SWE reported by the two reference datasets at various spatial and temporal scales and explore how the choice of reference dataset affects the accuracy assessment and overall performance ranking of the products. This analysis assesses the feasibility of developing a combined (snow course + airborne gamma) continental-scale reference dataset, both for benchmarking the performance of gridded
SWE products and other hydroclimate applications.



## 2 Data

### 2.1 Gridded datasets

Fourteen gridded SWE products were validated in this study (Table 1). Products include those which utilize EO data, coupled land-atmosphere reanalysis (with and without separate snow models and/or data assimilation), snow models of

varying complexity driven by reanalysis data, and data assimilation schemes. Some products (e.g. ERA5, JRA-55, Snow CCI and U Arizona) assimilate in situ snow depth measurements while others (e.g. ERA5-Land, MERRA2 and GLDASv2.2) do not. Products are described in the references listed in Table 1 except for ERA5-Snow which is an offline run of ERA5 without the assimilation of the IMS snow extent product to remove a temporal discontinuity associated with the introduction of the assimilation of IMS data in 2004 (Mortimer et al., 2020; Ochi et al., 2023). All products cover the northern

hemisphere except the U. Arizona dataset, which is limited to the Coterminous US (CONUS). The current product suite includes datasets which were part a previous evaluation reported in Mortimer et al. (2020), extended now with updated product versions and entirely new products.



**Table 1: Overview of the evaluated gridded SWE products.**

| Product | Abbr. | Period | Grid | Method | Snow Assim. | Reference |
|---|---|---|---|---|---|---|
| JAXA AMSR2 | JX | 2014 – 2018 | 12.5 km | Standalone passive microwave | None | Kelly et al. (2009) |
| Snow CCI CDR v1 | C1 | 1980 – 2018 | 0.25° | Passive microwave + snow depth assimilation | In situ snow depth | Luojus et al. (2021) |
| Snow CCI CDR v2 | C2 | 1980 – 2020 | 0.1° | Same as C1 except grid spacing and variable snow density | In situ snow depth | |
| Brown-ERA5 | BE | 1981 – 2018 | 0.25° | Temperature-index snow model + ERA5 forcing | None | Brown et al. (2003) Elias-Chereque et al. (submitted) |
| Brown-JRA55 | BJ | 1981 – 2018 | 1.25° | Temperature-index snow model + JRA55 forcing | None | |
| Brown-MERRA2 | BM | 1981 – 2018 | 0.5° x 0.625° | Temperature-index snow model + MERRA2 forcing | None | |
| Crocus-ERA5 | Cr5 | 1980 – 2021 | 0.25° | Crocus snow model + ERA5 forcing | None | bertrand.decharme@meteo.fr |
| ERA5-Land | EL | 1980 – 2018 | 0.1° | Reanalysis (HTESSEL LSM) | None | Muñoz-Sabater et al. (2021) |
| ERA5 | E5 | 1980 – 2018 | 0.25° | Reanalysis (HTESSEL LSM) | In situ snow depth + IMS | Hersbach et al. (2020) |
| ERA5-Snow | ES | 1980 – 2018 | 0.25° | Reanalysis (HTESSEL LSM) | In situ snow depth | patricia.rosnay@ecmwf.int |
| GLDASv2.2 | G2 | 2003 – 2018 | 0.25° | Reanalysis (Catchment LSM) | None | Rodell et al. (2004) |
| JRA-55 | JR | 1980 – 2018 | 55 km | Reanalysis (Simple Biosphere LSM) | In situ snow depth + PMW | Kobayashi et al. (2015) |
| MERRA2 | M2 | 1980 – 2018 | 0.5° x 0.625° | Reanalysis (Catchment LSM) | None | Gelaro et al. (2017) |
| U. Arizona | UA | 1981 – 2017 | 4 km | Data assimilation: surface snow observations + PRISM temperature and precipitation | In situ snow depth and SWE | Zeng et al. (2018) |



## 2.2 Reference datasets

### 2.2.1 Snow course SWE

Snow courses, also known as snow transects, consist of manual gravimetric snow measurements made at multiple locations along a predefined transect averaged together to obtain a single SWE value on a given date (WMO, 2018). In Canada, measurements are typically conducted once or twice per month during the snow season, although some sites are only sampled near the timing of peak SWE, and measurements are very sparse across the Arctic (Vionnet et al., 2021). In the

United States, measurements start in late December in high elevation areas of the west and throughout Alaska and after 1 January in the northeast. Measurement uncertainty for various snow samplers ranges from ~3% to 13% (Table 2 in Dixon and Boon, 2012 and references therein; López-Moreno, 2020). The snow course measurements used in this study (Table 2, Fig. 1) are independent of the data assimilated into any of the gridded SWE products with the exception of SnowCCI v2, which used an older Canadian dataset (Brown et al., 2019) for a dynamic density correction.

**Table 2: Reference data used in this study.**

| | Coverage | Data provider | Reference and data access |
|---|---|---|---|
| **Snow course** | Canada | CanSWE v3 - Environment and Climate Change Canada and partners | Vionnet et al. (2021) https://zenodo.org/record/5217044#.YdYEsllybb0 |
| | Western US and Alaska | U. S. Department of Agriculture Natural Resources Conservation Service (NRCS) | https://www.nrcs.usda.gov/wps/portal/wcc/home/snowClimateMonitoring/snowpack/ |
| | Northeast US | Northeast Regional Climate Centre | https://www.nrcc.cornell.edu/ |
| | | New Hampshire Department of Environmental Services – Dams | https://www.des.nh.gov/ |
| | | Maine Geological Survey | https://mgs-maine.opendata.arcgis.com/datasets/maine-snow-survey-data/explore |
| **Gamma** | US & transboundary Canadian watersheds | NOAA National Operational Hydrologic Remote Sensing Center (NOHRSC) | https://www.nohrsc.noaa.gov/snowsurvey/ |

### 2.2.2 Airborne gamma SWE

The attenuation of gamma radiation by the water mass of the snowpack (liquid or solid phase) can be related to SWE provided the background soil moisture is properly accounted for (Carroll, 2001). The US National Oceanographic and

Atmospheric Administration's (NOAA) National Operational Hydrologic Remote Sensing Center (NOHRSC) snow survey





program (https://www.nohrsc.noaa.gov/snowsurvey/) has been using airborne gamma measurements to estimate SWE operationally since 1979 (Carroll, 2001). Flights are conducted to measure gamma radiation when the ground is snow-free (background attenuation by soil moisture only) and again when the ground is snow-covered (attenuation by soil moisture and the snowpack). The operational equations used to relate gamma radiation to SWE are described in Carroll (2001). The
detection limit for this method is ~1000 mm SWE.

The NOHRSC snow survey network (Table 2, Fig. 1) consists of approximately 2,400 flight lines in 25 US states and seven Canadian provinces (Carroll, 2001). SWE is reported as an aerial average for each flight line, which is typically 15–20 km long across a 300 m wide footprint. Flights are conducted near the peak of the snow accumulation season and during melt when SWE information is critically needed for water supply outlook and flood forecasting, typically February through April
depending on the location. Spatial coverage of this dataset has varied over the years especially over the western US, and flights over Alaska only began in 2003. Accuracy of these data, determined from comparisons with coincident ground-based snow observations during specific field campaigns, is 4–10% in prairie environments across a SWE range of 20–150 mm (Carroll and Schaake, 1983) and 23 mm in densely forested terrain across a SWE range of 20-480 mm (Carroll and Vose, 1984).

## 3 Methods

### 3.1 Evaluation of gridded datasets

The analysis period for each product is listed in Table 1 and generally covers 1980–2020. Our analysis was restricted to February through April, when the ratio of snow courses to gamma SWE is most consistent (Fig 1a). Validation statistics (bias, unbiased root mean squared error - uRMSE, and correlation) were computed for North America, except U. Arizona
(CONUS-only). Our uRMSE estimate is defined as the square root of the mean square error minus the squared bias. Statistics are computed for all non-zero SWE ≤ 1000 mm (both reference and product SWE must be ≤ 1000 mm for inclusion) as well as for a subset of cases when the reference SWE is ≤ 200 mm (see Sect. 4.4). The upper (1000 mm) threshold is consistent with the maximum detection limit of the airborne gamma SWE method.

Reference SWE was matched up in space and time with gridded SWE at the native product resolution. To reduce errors from
mismatched water and ice masks, only sites with coincident product estimates from two thirds of the products evaluated were retained. For gamma SWE, we used the midpoint of each flight line for geolocation, which differs slightly from Cho et al. (2019; 2020) and Tuttle et al. (2018) who weighted the average of the gamma SWE footprint (using a fixed diameter of 330 meters assigned to each flight line) contained within each product grid cell. We found that both methods produced similar results, so we used the flight line midpoint for simplicity.

To reduce oversampling in areas with spatially dense networks, the reference data shown in Fig. 1b were resampled to a 100 km spacing (Fig 1c). Snow course and gamma SWE were considered separately, and mountain measurements were separated from non-mountain. To avoid oversampling specific grid cells, we first aggregated reference sites within the same product





grid cell (at the native resolution of the product grid) before aggregating to the 100 km spacing. Sensitivity analysis of various spatial aggregation distances between 50 and 200 km showed little impact of aggregation distance. We selected 100

km as a compromise between sample size and spatial distribution.

Due to the well documented challenges in estimating and validating mountain SWE at coarse resolutions, (Wrzesien et al., 2019) we also computed metrics separately for mountain and non-mountain reference data. Mountain sites are defined as those intersecting the Global Mountain Biodiversity Assessment (GMBA) Mountain Inventory v2 (Snethlage et al., 2022; https://www.earthenv.org/mountains) with a 25 km buffer or with a 2° slope mask derived from the GETASSE30 DEM. The

25 km buffer was added to the GMBA mountain mask to avoid contamination of product grid cells with fractional mountain terrain. SnowCCI products were excluded from our analysis of mountain regions because SWE is not provided across a complex terrain mask applied to those datasets.

### 3.2 Diagnosing the impact of reference data characteristics

We evaluated how differences in measurement method (snow course versus gamma), the spatiotemporal distribution of the

reference data, SWE magnitude of the reference datasets, and gridded product versus reference data elevation biases impact both absolute and relative product accuracies. For each of these covariates, a difference of means test (two-sided independent student t-test) was applied to determine whether the mean product metrics calculated using snow courses are different from those obtained with airborne gamma, using a significance level of 95%. Consistency in product rankings was assessed with the Spearman rank correlation coefficient.

### 3.2.1 Reference data measurement method

To investigate how the reference measurement approach impacts the reference SWE value we quantified the agreement between snow courses and gamma data at various spatial (5 km, 10 km, 25 km, and 50 km) separations and temporal (0, ± 3, ± 7 and ± 10 day) lags, separately for mountain and non-mountain regions. This analysis was conducted on the unaggregated reference data (Fig 1b). The spatial separation of measurements in the two datasets was taken as the linear distance between

the snow course location and the gamma flight line midpoint. Using the measurement accuracies described in Sect. 2, we assigned a 10% uncertainty to both reference datasets. Addition in quadrature of these independent uncertainties yields a combined baseline uncertainty of ~14% for the airborne gamma – snow course comparison. This value is likely an underestimate and does not consider issues of spatial representativeness and spatial scale which cannot be quantified with the available data used in this study, nor does it consider operator error in the case of snow courses (López-Moreno et al., 2020).

### 3.2.2 Spatial distribution of the reference data

To evaluate the impact of differences in spatial coverage beyond the scale of a typical grid cell (Table 1), we generated matched reference data subsets composed of snow course sites having at least one gamma site within a specified linear distance and vice versa. We tested various distances between 25 km and 500 km, separately for mountain and non-mountain



sites (i.e. reference sites where both are classified as either mountain or non-mountain). For each spatial separation distance,
the matched snow course and gamma reference data were each spatially aggregated to 100 km as described in Sect. 3.1. In
this analysis of spatial distribution, no restriction was placed on temporal separation of the reference data, meaning the
reference sites can be from any date during the analysis period. In this way, spatial distribution serves as a proxy for similar
landcover types.

### 3.2.3 SWE magnitude

To evaluate the impact of SWE magnitude on relative and absolute product accuracies, we calculated validation metrics for
sequential 50 mm SWE bins based on the February-April climatological mean of the spatially aggregated reference sites
(Sect. 3.1, Fig. 1c) having at least five observations during the study period.

### 3.2.4 Elevation bias

In mountain regions, large changes in elevation over short distances are common. At the kilometre scale, SWE generally
increases with elevation up to a certain point, after which it decreases due to wind redistribution particularly above treeline
(Grünewald et al., 2014; Kirchner et al., 2014). If reference measurements are consistently collected at higher (lower)
elevations relative to a product grid cell centroid, we might expect them to have more (less) SWE compared to the grid cell
average. To understand the impact of elevation bias on SWE validation statistics, we compared the elevation of product grid
centroids with reference data elevations. We used the model surface geopotential height converted to metres or in the case of
Crocus and U Arizona, the DEM used by the model. For EO products which do not rely on any type of elevation
information, we used the GLOB30 DEM re-gridded to the native product grid. Reference data elevations were screened for
outliers using the USGS 30-metre NED1 DEM (NED2 60m DEM for Alaska) (Gesch et al., 2018) and sites with metadata
elevations > |1000| m from the NED1 DEM were removed. Reference sites without accompanying metadata elevations (16
gamma sites, 33 snow course sites) were assigned the intersecting USGS NED DEM elevation. We computed validation
statistics for sequential 100 m elevation-bias bins.

## 4 Results

### 4.1 Overall gridded product performance

The relationship between gridded SWE products and reference data over the full spatial domain (using the aggregated
reference data as shown in Fig. 1c) is shown in Fig. 2. There are clear differences in performance among the products. U.
Arizona outperforms all products regardless of the reference dataset, but the dataset domain, and hence the validation
statistics, are limited to CONUS. Considering the entire North American domain, ERA5-Land, ERA5, ERA5-Snow and
Crocus-ERA5 consistently rank among the top half of the products evaluated, albeit with some differences according to the
metric and reference dataset. The Brown temperature index model products, despite employing relatively simple



formulations for snow processes, also have reasonable performance, although the JRA55 forcing results in poor correlations against snow courses (Fig. 2). A strength of the U. Arizona, ERA5-Land and Crocus-ERA5 products is that they show good performance across the full range of reference SWE values (Fig. 2). While ERA5 and ERA5-Snow have strong correlations against both reference datasets, these products suffer from larger biases in high SWE regions because their maximum snow depth is fixed at 1.4 m (corresponding to ~500 mm SWE depending on the snow density) to prevent excessive snow accumulation at high elevations and latitudes (P. de Rosnay, pers com). The JAXA-AMSRE, JRA-55, MERRA2 and GLDASv2.2 products exhibit the weakest performance. SWE estimates from these products have very weak statistical relationships to observed SWE. This makes them unsuitable to discern the impact of covariates (i.e., sampling methodology, spatiotemporal distribution, SWE magnitude, product-reference elevation bias as described in Sect. 3.2.4) on the differing product statistics obtained with snow courses and airborne gamma SWE, so they are excluded from this analysis in Sect. 4.2 through 4.5. The best performing EO-based products (Snow CCI) saturate when reference SWE exceeds ~250 mm (Fig. 2) which is an important consideration for the appropriate use of these datasets.

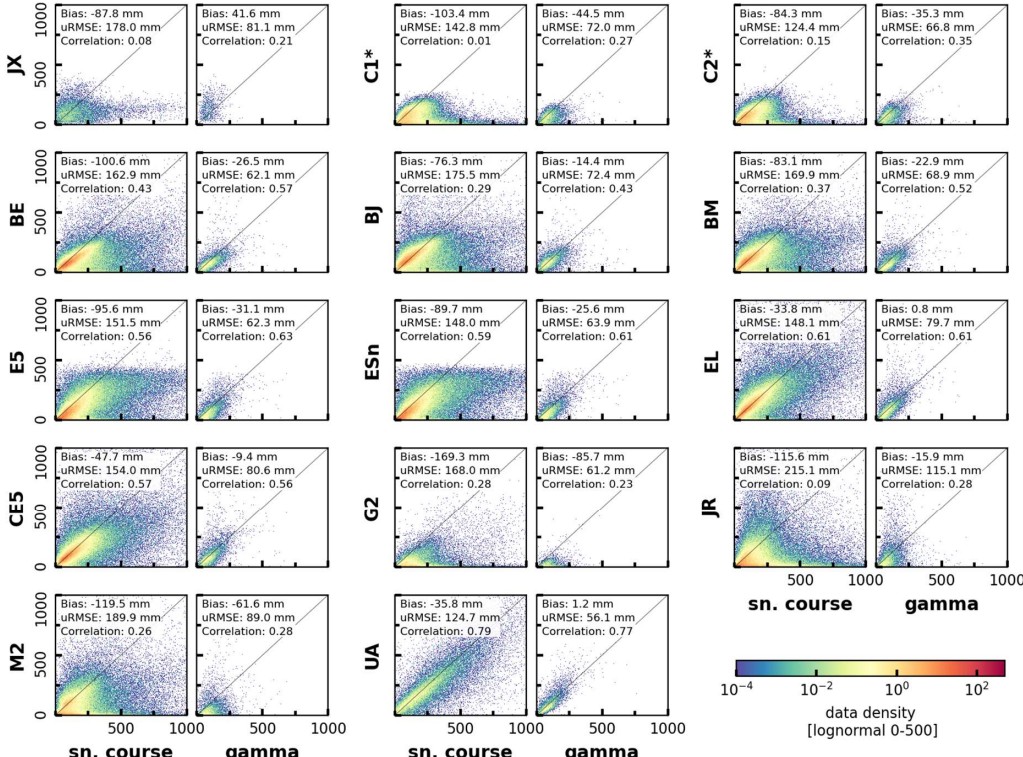





**Figure 2: Product vs reference SWE density scatter for measurements > 0 and ≤ 1000 mm during February – April. See Table 1 for product names, descriptions, and time periods. Note that Snow CCI excludes areas of complex terrain, U Arizona is limited to CONUS, and JAXA-AMSR2 (2014-) and GLDASv2.2 (2003-) are limited temporally.**

The validation statistics shown in Fig. 2 provide benchmark information on the performance of currently available gridded SWE products over North America, building on an earlier assessment of a previous generation of products (Mortimer et al., 2020). For all products, the bias magnitude and uRMSE are larger using the snow course reference dataset as compared to the airborne gamma reference dataset. This difference is due in large part to the higher proportion of mountain snow course sites for which validation statistics are poor (Fig. 3): over 50% of the snow course data are located in mountain areas

compared to just over 30% of gamma the data. Product performance is considerably worse in mountain compared to non-mountain regions (Fig. 3), but the discrepancy is larger when snow courses are used as the reference data compared to gamma SWE. In the following sections, we evaluate how differences in measurement method, spatiotemporal distribution, reference SWE magnitude, and gridded product elevation biases impact both absolute and relative product statistics. A subset of products with coverage of both mountain and non-mountain areas and a reasonable relationship with reference

SWE as determined from Fig. 2 – Crocus-ERA5, Brown-ERA5, Brown-MERRA2, Brown-JRA55, ERA5, ERA5-Land, ERA5-Snow, U. Arizona – are used for this analysis.

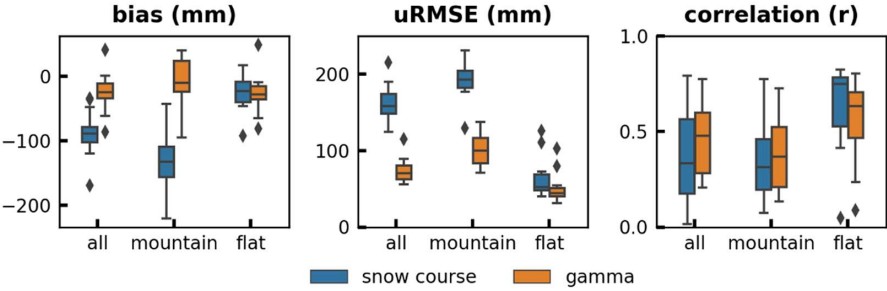

**Figure 3: Boxplot of statistical performance for products listed in Table 1 computed separately for snow courses versus gamma**
**SWE, and mountain versus non-mountain ("flat") regions across the full spatial domain.**

**4.2 Reference data measurement method**

Comparisons of snow course versus gamma reference SWE at various spatial separation distances (Fig. 4a) and temporal lags (Fig. 4b) demonstrate poor agreement in the mountain regions even at short distances and no temporal lag (Fig. 4a, bottom row). Depending on spatial separation distance and temporal lag, the mean difference between gamma and snow

course measurements in mountain regions varies from 35–55% of the mean reference SWE and the correlation of the two reference measurements drops markedly with increasing distance. The agreement between snow course and gamma reference SWE is much stronger in non-mountain regions. When constrained to the same dates, the mean difference between





gamma and snow courses is approximately 20% across all spatial distances tested (Fig. 4a top row shows mean bias ~20mm and mean reference SWE ~100mm). Relaxing the temporal constraint (Fig. 4b) allows for many more paired measurements

such that the mean difference in non-mountain regions drops to less than 12% of the mean reference SWE for temporal lags larger than zero (Fig. 4b, top row) which is within our 14% baseline uncertainty estimate for this comparison (Sect. 3.2.1). In addition, for all separation distances and temporal lags, the non-mountain reference SWE values are reasonably correlated with one another: generally above 0.7 although their correlation drops slightly at the longest temporal lags. Therefore, we conclude that in non-mountain terrain, the reference dataset measurement method does not result in detectable differences in

the determination of product performance, up to the spatial (50 km) and temporal (10 d) lags evaluated. We note here that the agreement of the reference datasets in non-mountain regions as evaluated up to 50 km is comparable to the grid spacing of the majority of SWE products considered.  However, because the majority (> 90%) of the matched data are located in either the forested Northeastern US or the NA L1 Great Plains Ecoregion (https://www.epa.gov/eco-research/ecoregions-north-america), we are unable to extrapolate this result directly to all North American non-mountainous regions.




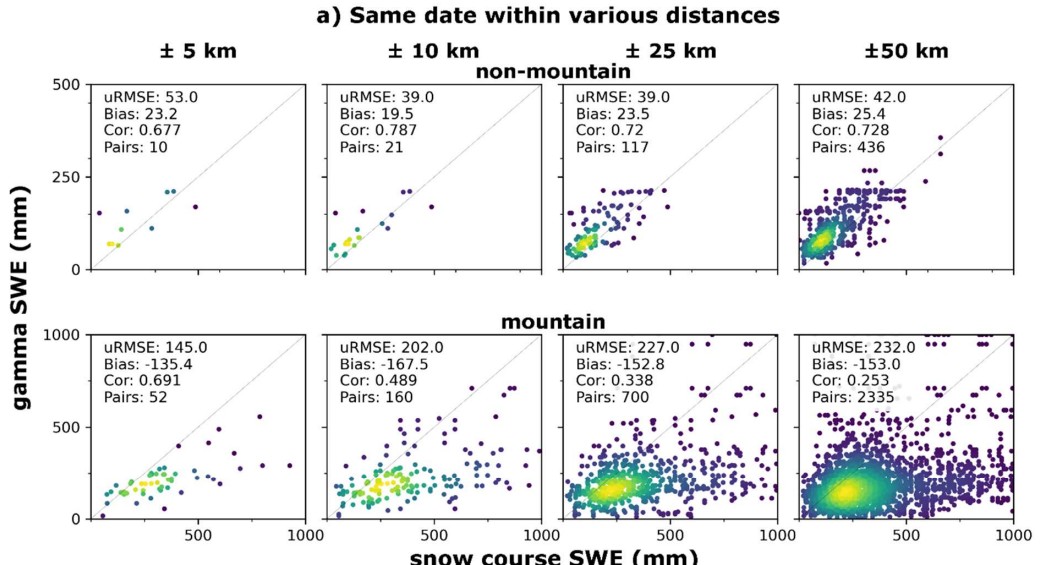

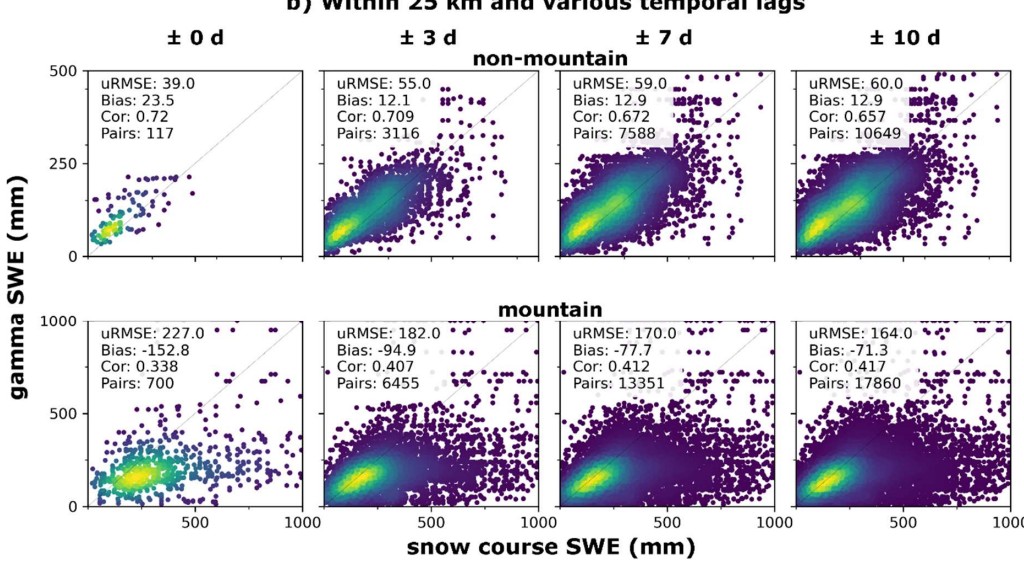

**Figure 4: Snow course versus gamma measurements. (a) measurements on the same date within 5, 10, 25 and 50 km for SWE > 0 and ≤ 1000 mm. (b) measurements within 25 km on the same date and within 3, 7 and 10 days.**



### 4.2 Reference data spatial distribution

Having directly compared the two sources of reference SWE above, in the following sections we provide comparisons of product performance metrics calculated using one reference dataset or the other and demonstrate under what conditions and to what extent they agree. This analysis remains restricted to the Crocus-ERA5, Brown-ERA5, Brown-MERRA2, Brown-JRA55, ERA5, ERA5-Land, ERA5-Snow and U. Arizona products. First, we reassess differences for this product suite over the full spatial and temporal domain, separately for mountain and non-mountain areas. Over the full domain, there is

reasonable agreement in relative product rankings of all three metrics in both mountain and non-mountain regions (expressed by Spearman correlation coefficients, Fig. 5). However, there is poor agreement on the absolute bias and uRMSE magnitudes, especially in mountain regions: the grey circles do not lie on the 1:1 line and the mean difference in product bias and uRMSE computed with either reference dataset is larger than the uncertainty envelope attributed to measurement error (Fig. S1, orange dots). Further, the mean product statistics calculated with either reference dataset are distinct from each

other (p < 0.05) except for the bias in non-mountain regions and the correlation in mountain regions (Fig. S1, blue dots).

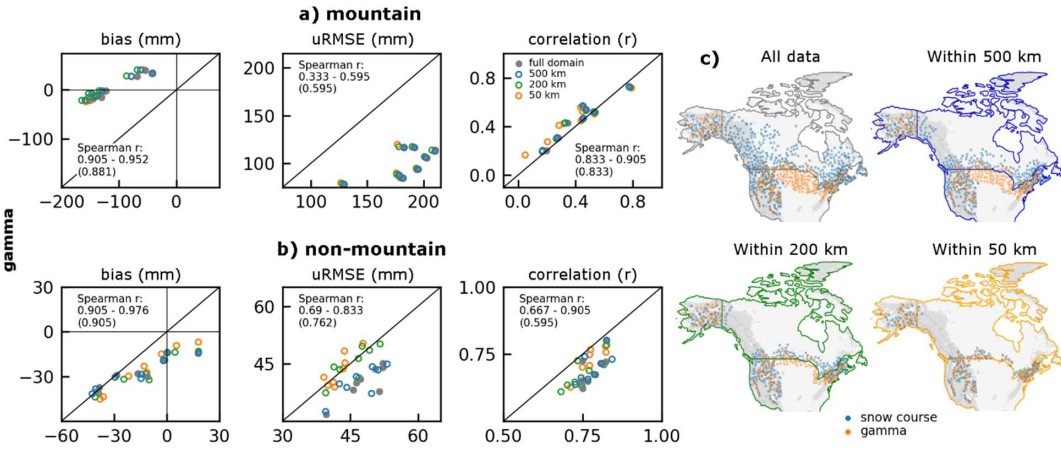

**Figure 5: Product performance metrics computed with gamma SWE vs snow courses for the full spatial domain (solid grey circles) and for reference data spatial subsets (c) defined by the linear distance between the gamma and snow course sites for (a) mountain and (b) non-mountain regions. For example, 200 km refers to metrics calculated using only the subset of gamma (snow course)**

**reference data having a snow course (gamma) site within 200 km. Each dot represents one of the products Brown-ERA5, Brown-JRA55, Brown-MERRA2, ERA5, ERA5Snow, ERA5-Land, Crocus-ERA5, and U. Arizona. Spearman correlation coefficients, which assess agreement in product rankings calculated with snow courses versus gamma SWE, are summarized by the range across the spatial lags tested and for the full domain (parentheses).**



The statistical differences in metrics assessed from the snow course versus gamma reference datasets as described above may stem from differences in where each reference dataset has coverage (Fig. 1). Gamma SWE is limited to the US and southern Canada, and so misses the high SWE areas of the boreal forest (e.g., northern Quebec, Canada) sampled by snow courses. Roughly one third of the gamma data are in the NA L1 Great Plains Ecoregion which has limited (< 2.4%) snow course data for the paired analysis as described in Sect. 4.2. Conversely, 12% of the snow course data are in arctic regions

(NA L1 Ecoregions – Arctic Cordillera, Taiga, Tundra, Hudson Plain) compared to just 3% of the gamma data. Unless all products perform equally well across all landcover types (which is very unlikely), these spatial differences are likely to result in differing product accuracies calculated for snow courses versus airborne gamma. To examine how sensitive the assessed product performance is to differences in spatial coverage of the reference data, performance metrics using only reference sites within 25 km to 500 km (Sect. 3.2.2) of each other are also shown in Fig. 5 (for display purposes, only 50 km, 200 km

and 500 km are shown).

Spatially restricting the reference data had a minor impact on the agreement in product ranking (Spearman correlation coefficients, Fig. 5). In mountain regions, spatially restricting the analysis domain resulted in minimal change in product metrics (Fig. 5a), and thus no improvement in the discrepancy in product metrics according to choice of reference dataset (Fig. S1). Spatially restricting the reference data did, however, alter the uRMSE in non-mountain regions (Fig. 5b). At

smaller spatial scales (< 300 km) the discrepancy in product uRMSE decreases and the values calculated with either snow courses or airborne gamma reference data become statistically indistinguishable from one another (Fig. S1).

The fact that the two reference datasets still yield different performance metrics in mountain regions despite the spatial restrictions applied above suggests that (i) our spatial domains are not sufficiently restrictive or that we need to also consider the temporal domain or that (ii) additional factors such as SWE magnitude or elevation bias have a greater impact on

reference dataset performance agreement than does spatial distribution. To test the impact of temporal distribution on absolute and relative product statistics we further restricted the reference dataset to only include nearby (within 200 km) measurements collected within ten days of each other (Fig. 6). The temporal distance was informed by our direct comparison of reference SWE (Sect. 4.2, Fig. 4) which showed little impact on agreement up to ten days. For the spatial distance we considered sample size and the distance at which non-mountain statistics converged. Specifically, in non-mountain areas

there was no difference in mean product uRMSE computed with either reference dataset when constrained to sites within 250 km of each other, and there is lower agreement in non-mountain product ranking (bias and correlation) at spatial separation distances > ~150–200 km (Fig. S2). This spatial subset comprises most of the mountain observations (> 95% of gamma data, ~75% of snow course data) but only around half (~60% of gamma data, ~50% of snow course data) of the non-mountain data.



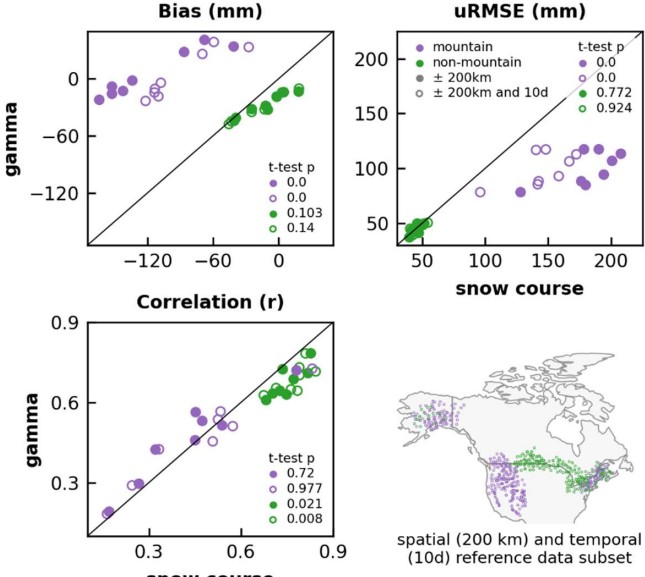


**Figure 6: Absolute value of difference in product metric computed with snow courses vs gamma SWE sites within 200 km of each other for a subset of reference measurements collected within seven days of each other versus those collected at any time during Feb.–April 1980–2020. Each dot represents one of the products: Brown-ERA5, Brown-JRA55, Brown-MERRA2, ERA5, ERA5Snow, ERA5-Land, Crocus-ERA5, and U. Arizona, in mountain (purple) and non-mountain (green) terrain. p-values from**

**two-sided independent student t-test (Sect. 3.2) comparing difference in mean product statistic computed with snow course and with airborne gamma. When p < 0.05 the ensemble mean metrics computed using each choice of reference data are statistically distinguishable at 95% confidence. Lower right map shows reference data locations of the spatially and temporally constrained subset (not separated by reference data type for display purposes).**

Temporally restricting nearby reference measurements reduced the mean difference in mountain [product] bias and uRMSE computed with the two reference datasets by over 25% (the hollow purple circles in Fig. 6 move closer to the 1:1 line compared to the solid purple dots), but these metrics remain statistically larger and distinct (p < 0.002) when computed against snow courses rather than airborne gamma. These differences are expected given the discrepancy in observed reference values at the measurement scale (Fig. 4). In non-mountain regions, agreement of the spatially restricted dataset was

already strong, and the temporal restriction did not result in any further improvement (Fig. 6). In the next sections we use this coincident reference data subset (200 km and ± 10 d) to explore the impact of SWE magnitude and elevation bias on absolute and relative product accuracies. This spatial and temporal subset comprises less than 20% of the original dataset so reference to the full domain is made where appropriate.



### 4.3 SWE magnitude

The relationship between product and reference SWE deteriorates for most products at higher SWE magnitudes (Fig. 2). In some cases, there is a clear contribution to this deterioration by thresholds which are applied to avoid excessive snow accumulation (i.e. ERA5 and ERA5-Snow, Sect. 4.1) or, in the case of EO approaches, as the physical retrieval basis is no longer applicable (Chang et al., 1982, 1987; Luojus et al., 2021). In mountain areas, despite having similar elevation

distributions, snow courses sample a much larger range of SWE than does airborne gamma, whether evaluating the complete spatial and temporal domain or the coincident subset (Fig. 7). In non-mountain areas, mean SWE observed by snow courses and gamma is comparable over the coincident reference subset but is slightly higher for snow courses over the full domain due to high SWE sites in the northern boreal forest which are not sampled by airborne gamma.



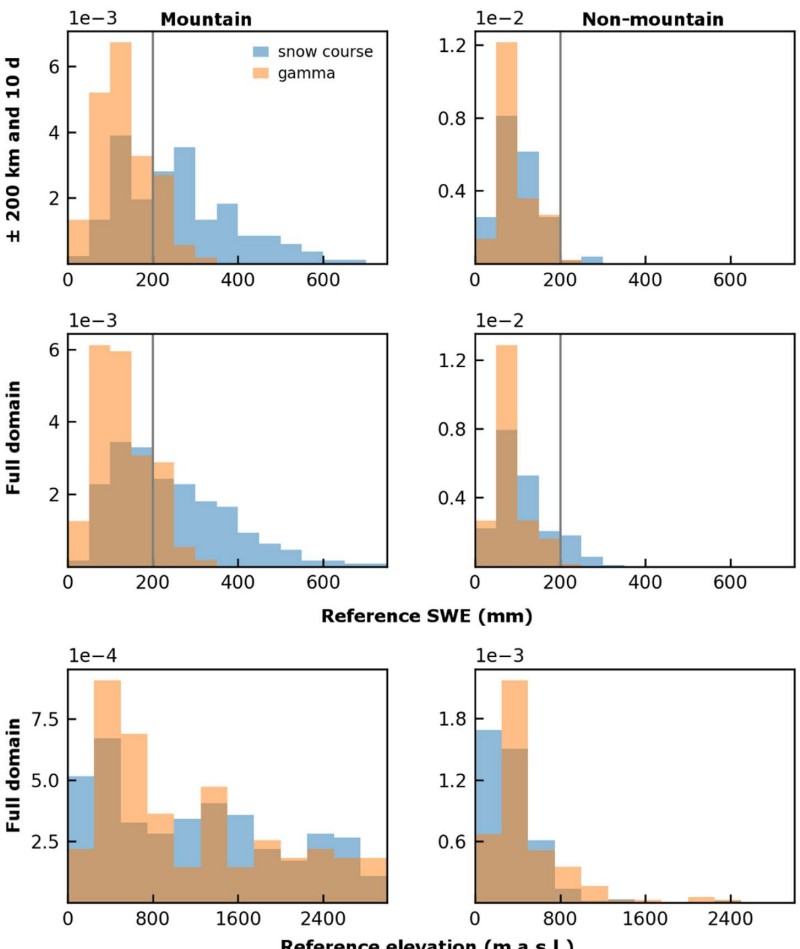

**Figure 7: Reference SWE (top two rows) and elevation (bottom row) distribution for spatially and temporally restricted subset (top row) and the full domain (bottom rows) for mountain (left) and non-mountain (right). The spatial and temporal subset (top row) is the same reference data used to calculate the product statistics shown in Fig. 6 (hollow dots).**



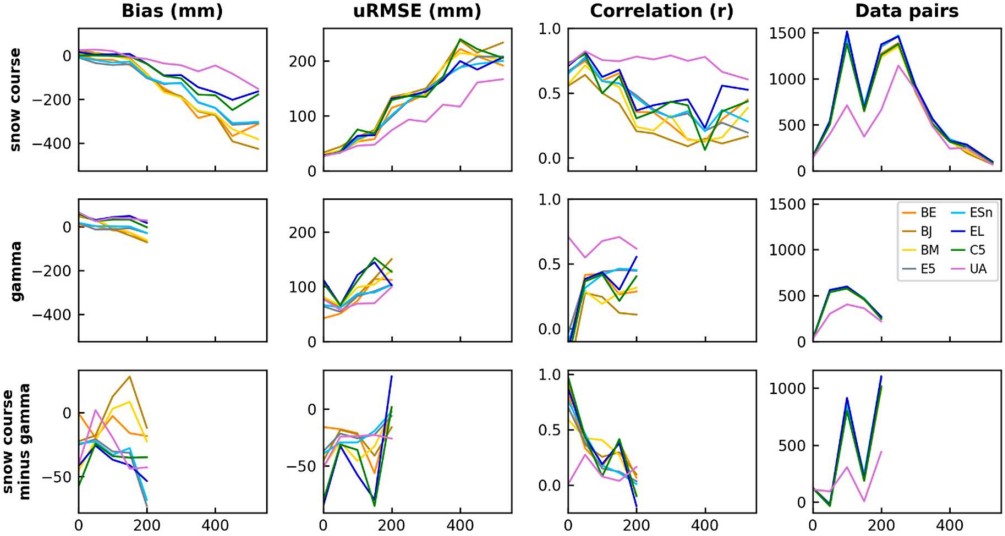

**Figure 8: Mountain product performance for sequential 50 mm SWE bins over the spatially and temporally restricted domain.**
**Bottom row shows the difference between snow course and gamma derived metrics for each product and bin. Product metrics shown are limited to 200 mm and 500 mm and below for gamma and snow course, respectively, due to limited number of data pairs above these thresholds.**

Bias and uRMSE increase with reference SWE magnitude for both mountain (Fig. 8) and non-mountain regions (not shown) whether computed using the full reference domain (Fig. S3) or the restricted one. The spread in product uRMSE and bias

values also increases with SWE magnitude; however, because airborne gamma observations are limited to moderate SWE values it fails to capture much of these inter-product differences. Correlations calculated using snow courses or gamma reference SWE are fairly stable between 50 and 150 mm; those calculated using snow courses drop sharply above the 150 mm bin except for U. Arizona, consistent with Fig. 2 which shows good agreement between reference and U Arizona SWE across the full SWE range.

Thresholding the coincident subset to the SWE domain consistently sampled by both reference datasets (200 mm and below as constrained by the gamma dataset) reduced the discrepancy in product metrics (bias and uRMSE) computed with snow courses and airborne gamma in mountain regions by over two thirds (Fig. 9a). The systematic negative product bias against snow courses observed over the full analysed mountain SWE range is removed when the high SWE sites responsible for much of this underestimation are excluded. Of course, restricting analysis in mountain regions to SWE < 200 mm is

unsatisfying since this represents very shallow mountain snow conditions. The uRMSE is reduced for both reference datasets but the magnitude of improvement is considerably larger for the snow course dataset improving the inter-reference-dataset



agreement (Fig. 9a, hollow purple p = 0.04). Restricting the reference dataset to sites with climatological SWE ≤ 200 mm had negligible impact on non-mountain product metrics (and so not shown on Fig. 9a) because the non-mountain reference SWE distributions are similar and mostly below this threshold (Fig. 7).

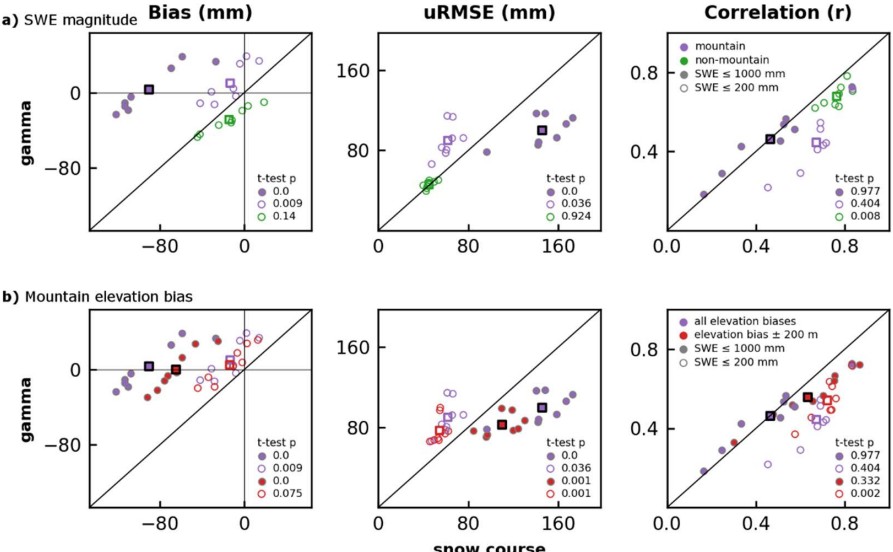


**Figure 9: Sensitivity of product metrics computed with gamma SWE versus snow courses to (a) SWE magnitude and (b) elevation bias for the spatially and temporally restricted domain. (a) Mountain (purple) and flat (green) metrics for SWE ≤ 1000 mm (solid circles) and SWE ≤ 200 mm (hollow circles). (b) Mountain product metrics for reference-product pairs with elevation biases <|200| m (red circles) and no elevation bias restriction (purple) for the full (solid) and restricted ≤ 200 mm (hollow) SWE domain. Each**

**dot represents one of the products Brown-ERA5, Brown-JRA55, Brown-MERRA2, ERA5, ERA5Snow, ERA5-Land, Crocus-ERA5, and U. Arizona. Squares represent the mean of the products. T-test p-values as in Fig. 6. Non-mountain SWE ≤ 1000 mm overlap almost exactly with SWE ≤ 200 mm and are not shown. Purple dots in (a) and (b) are the same.**

### 4.4 Elevation bias

A consistent high-level message is that products perform considerably worse in mountains compared to non-mountain areas.

Thresholding the analysis to moderate levels of SWE (≤ 200 mm) decreases the discrepancy in product statistics calculated with one reference dataset versus the other, but uRMSE and correlation are still worse in mountain compared to non-mountain regions (Fig. 9a). As outlined in Sect. 3.2.4, in mountain and complex terrain, the relationship between SWE and elevation can result in large SWE gradients over short distances (i.e. less than a single product grid cell). In these regions, systematic differences in elevation between reference measurements and the centroid of a product grid could, therefore,

produce validation errors that are a result of our approach rather than the products themselves.



To investigate the impact of elevation biases, we computed product metrics for sequential 100 m elevation difference bins (determined by the reference measurement location versus the centroid of a product grid) (Sect. 3.2.4). This analysis is restricted to mountain regions because elevation mismatches are smaller in non-mountain regions (< 10% of reference-product data pairs have elevation biases > |200| m). The mean metrics for Brown-ERA5, Brown-MERRA2, Brown-JRA55,

ERA5, ERA5Snow, ERA5-Land, Crocus-ERA5 and U. Arizona are shown in Fig. 10. Product SWE bias increases linearly with elevation difference and the minimum uRMSE occurs at or near zero elevation difference; correlations tend to be highest when elevation biases are smallest. The impact of elevation biases on product performance is reduced, but not eliminated, when SWE is restricted to 200 mm and below (Fig. 10, hollow circles). This is especially for snow courses and indicates that sites with large elevation biases also have moderate to high SWE. Surprisingly, we did not find a systematic

relationship between elevation bias and product grid spacing (e.g. larger and/or greater frequency of elevation bias for more coarsely gridded products).

Restricting the analyzed data pairs to those with elevation biases < |200| m (Fig. 9b, red) improves the mean bias and uRMSE across both the full and restricted SWE. However, the improvement is small compared to that achieved by SWE thresholding (Fig. 9a). The discrepancy in product metrics (bias and uRMSE) improves when the full SWE range (Fig. 9b,

solid circles) is restricted to data pairs with small elevation biases. When considering an already restricted set of pairs for which SWE ≤ 200 mm (Fig. 9b, hollow circles) there is limited to no additional improvement in the agreement in product metrics by further restricting the allowed elevation bias.

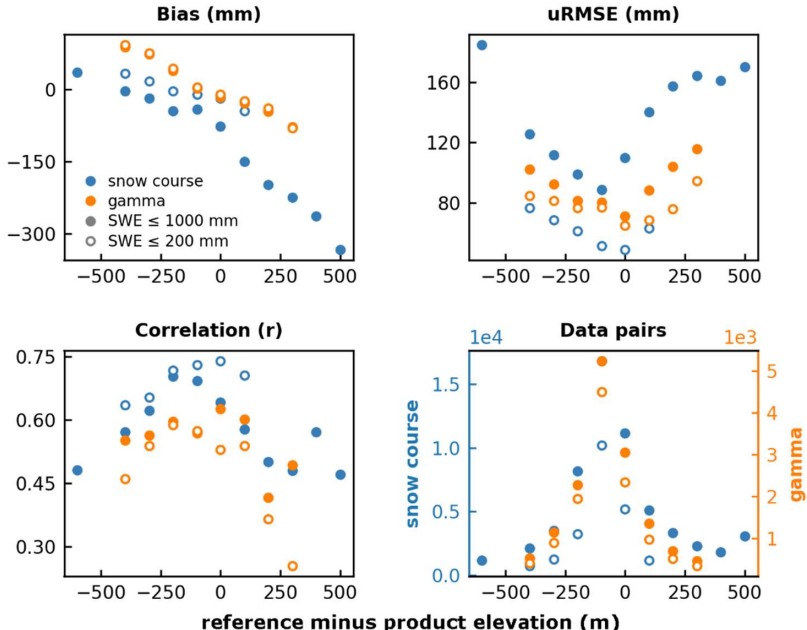



**Figure 10: Mean mountain bias, uRMSE and correlation and total number of reference–product data pairs for sequential 100 m**
**elevation bias bins (reference minus product elevation) for the products Brown-ERA5, Brown-JRA55, Brown-MERRA2, ERA5,**
      **ERA5Snow, ERA5-Land, Crocus-ERA5, and U. Arizona. Metrics are computed for the spatially and temporally restricted**
      **reference dataset (Sect. 4.3) over the full (≤ 100 mm) and restricted (≤ 200 mm) SWE ranges. X-axis labels indicate centre of 100 m**
      **elevation bin. Only bins with >1.5% of total data pairs are shown.**

### 5 Discussion

There are limited types of reference datasets available to evaluate gridded SWE products, with snow courses and airborne
      gamma data providing the most appropriate options. Our analysis shows that the choice of reference dataset has little impact
      on the general assessment of relative product performance (Fig. 2), but has a large impact on the magnitude of the statistics
      calculated in mountain areas.

#### 5.1 Non-mountain regions

There is no measurable difference in SWE measured by gamma or snow courses in non-mountain terrain up to the scale of
      most gridded products evaluated (Fig. 4). This result suggests that the reference data are sampling the true SWE field at
      spatial and temporal scales less than its intrinsic variability and hence the reference data SWE estimates are representative at
      scales appropriate for gridded SWE evaluation in non-mountain areas. When constrained spatially, which serves as a proxy
      for landcover type, validation statistics from either of the two reference datasets are comparable (Fig. 5). The fact that
validation statistics from either of the two independent reference datasets are comparable in non-mountain regions when
      evaluated over similar environments demonstrates that we can robustly validate gridded SWE estimates in such regions.
      Together, the strong agreement in reference SWE and consistent accuracies suggests that we can confidently use these two
      reference datasets in concert.

#### 5.2 Mountain regions

In mountain areas, challenges remain surrounding the estimation and evaluation of coarse-resolution gridded SWE products.
      The U Arizona SWE product demonstrates that strong performance in mountain areas is possible if observations from dense
      national in situ networks (SNOTEL and COOP) are combined with downscaled temperature and precipitation data (PRISM)
      at a fine spatial resolution (Zeng et al., 2018). Aside from this product, a consistent high-level message from our analysis is
      that products perform considerably worse in mountain compared to non-mountain areas. This is a well documented issue
(Fang et al., 2022 and references therein; Kim et al., 2021; Liu et al., 2022; Snauffer et al., 2016; Terzago et al., 2017;
      Wrzesien et al., 2019). However, our analysis also shows that the choice of reference data may also contribute to poorer
      product performances, as demonstrated by the large discrepancy in product metrics computed with the two reference datasets
      in coincident mountain areas (Fig. 6).



In mountain areas, SWE magnitude has the largest impact on product accuracies and their agreement according to the choice

of reference dataset (Fig. 9a). Elevation bias and the spatiotemporal distribution of the reference datasets have comparable impacts, both of which are smaller than that of SWE magnitude. High SWE observations also tend to have large elevation biases, so the order in which the elevation bias and SWE thresholds are applied influences their relative impact.

The systematically higher SWE measured by snow courses in mountain areas (compared to airborne gamma) (Fig. 7), even over short distances (5km) (Fig. 4b), translates in measurable differences in product bias and uRMSE (Fig. 3, Fig. 5a). It is

only possible to obtain reasonable reference-dataset agreement in product these metrics by restricting the analysis to sites with climatological mean SWE of 200 mm and below (Fig. 9a). Such a restriction is unrealistic as it omits a majority of the SWE range observed by snow courses, at which differences in product performance are greatest. Airborne gamma is known to underestimate SWE in areas with high snowpack spatial variability (Cork and Loijens, 1980; Cho et al., 2019; Carroll and Carroll, 1989), commonly caused by drifting snow or complex terrain.  Considering the full spatiotemporal domain, despite

having similar elevation distributions to snow courses, airborne gamma SWE estimates have a maximum of ~350 mm (Fig. 7), which best represents shallow mountain snow conditions which is an important consideration for the appropriate use of airborne gamma SWE estimates in these regions.

### 5.2 Towards a combined North American reference SWE dataset

Across North America, snow course and airborne gamma networks have largely complementary spatial coverage. Creating a

combined reference dataset using both of these sources contributes to a fuller picture of gridded SWE dataset performance. Given the strong agreement in reference SWE and their derived product accuracies in non-mountain areas, we are confident in pooling all snow course and gamma SWE observations together and computing a single product accuracy from these pooled data. As there is limited overlap between the datasets in the non-mountain domain outside of the Northeastern US, weighting as a function of footprint size or spatiotemporal sampling density is not necessary.

In mountain regions, the decision on whether and how to combine the reference data is less straightforward because of the lack of agreement between snow course measurements and gamma-derived estimates of SWE (e.g. Fig. 4b). Reasonable alignment in mountain SWE product metrics from the two reference datasets only occurs when constraints on similar locations, dates, and climatological SWE are applied (Fig. 9). The higher spatial and temporal density of the snow course dataset (Fig. 1) will bias any combined dataset towards snow courses, and we considered evenly weighting snow courses and

airborne gamma in mountain regions to address this. However, Fig. 7 suggests the gamma SWE distribution is shifted toward lower values in mountain areas, so evenly weighting snow courses and gamma SWE would artificially reduce the relative peak SWE. Given these constraints we chose not to combine snow course and gamma SWE data in mountain areas. Instead, we consider these two reference datasets separately in mountain regions. The new combined reference dataset, illustrated in Fig. 11 (described in Sect. 3.1), will contribute to a comprehensive evaluation of Northern Hemisphere gridded

SWE products in the context of the European Space Agency 'Satellite Snow Product Intercomparison and Evaluation Exercise' (SnowPEx; Mudryk et al., submitted) beyond the statistical validation described in this study.





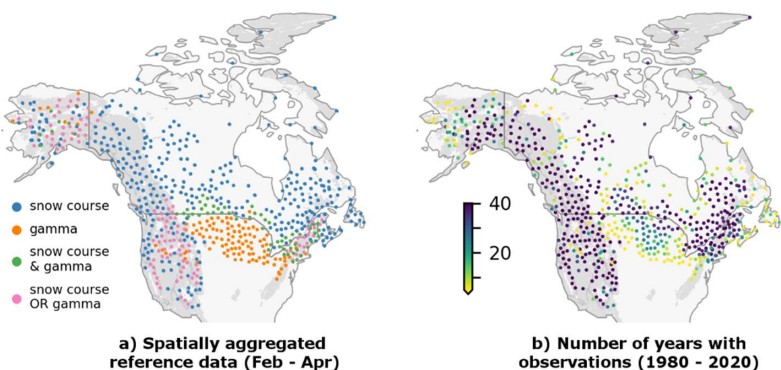

**Figure 11: Combined and spatially aggregated (100 km) in situ SWE dataset consisting of snow course and airborne gamma SWE measurements over north America for February through April 1980–2020. Grey shading indicates mountain regions.**

## 6 Conclusions

The choice of reference dataset has little impact on SWE product ranking but a large impact on the magnitude of validation statistics in mountain regions (Fig. 3). The strong agreement in non-mountain areas occurs because the reference SWE measured by gamma or snow courses agrees up to the scale of most gridded products evaluated (Fig. 4). In mountain regions the poor agreement in product statistics results primarily from the larger SWE range sampled by snow courses compared to gamma SWE (Fig. 7). Reasonable agreement in mountain product statistics was only achieved by restricting the reference data to similar dates, locations, climatological SWE, and elevation biases (Fig. 8). This approach is ultimately not appropriate, however, as it omits all but shallow-to-moderate mountain snowpacks indicating that targeted approaches are required to validate SWE products in mountain regions. Building on insights gained from our analysis of reference SWE agreement and of the impact of covariates on product accuracies (SWE magnitude and product-reference elevation offsets both impact absolute and relative product performance) we produced a combined spatially aggregated North American reference SWE dataset (Fig. 11) for non-mountain areas consisting of snow course and airborne gamma measurements.

**Data Availability**

The reference data are available using the links in Table 2 except for those from the Northeast Regional Climate Centre and New Hampshire Department of Environment which are available from the authors upon request. The combined reference dataset will become available by the time of publication. Gridded SWE product (Table 1) availability as listed below.



| Product Name | Availability/DOI |
| --- | --- |
| B-TIM-ERA5 | DOI to be available at time of publication |
| B-TIM-JRA55 | DOI to be available at time of publication |
| B-TIM-MERRA2 | DOI to be available at time of publication |
| Crocus-ERA5 | DOI to be available at time of publication |
| ERA5 | 10.24381/cds.adbb2d47 |
| ERA5-Snow | Available on request from patricia.rosnay@ecmwf.int |
| ERA5-Land | 10.24381/cds.e2161bac |
| GLDAS v2.2 [CLSM] | 10.5067/TXBMLX370XX8 |
| JRA-55 | https://jra.kishou.go.jp/ |
| MERRA2 | 10.5067/RKPHT8KC1Y1T |
| Snow_CCI v2 | 10.5285/4647cc9ad3c044439d6c643208d3c494 |
| Snow_CCI v1 | 10.5285/fa20aaa2060e40cabf5fedce7a9716d0 |
| JAXA-AMSR2 | Preliminary version provided as part of SnowPEx+. Available on request from rejkelly@uwaterloo.ca |

**Author Contributions**

CM, EC, LM and CD developed the analysis framework. CM and LM developed the code to calculate statistics and performed the analysis. CM prepared the manuscript with contributions from all co-authors.

**Competing Interests**

At least one of the (co-)authors is a member of the editorial board of The Cryosphere.

**Acknowledgements**

This work was initiated through the ESA-funded SnowPEx+ project.

**Data availability**

Reference data and gridded SWE products are available using the links in Tables 1 and 2 or on request from the authors where direct links are not available. The NOAA airborne gamma survey data are available at https://www.nohrsc.noaa.gov/snowsurvey/historical.html.



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
