# Peer review of "Use of multiple reference data sources to cross validate gridded snow water equivalent products over North America"

_EGUsphere, 2023_

## Author Comment (AC2)

**Response to Reviewer #1**

In this work, the authors utilized snow course and airborne gamma data from North America to comprehensively evaluate the performance of grid snow water equivalent products in both mountainous and non-mountainous areas at various spatial and temporal scales. Additionally, a combined reference SWE dataset for North America was produced. It's a challenging but promising endeavor. Here, I would provide some comments and suggestions for authors' consideration when revising the paper.

Comments:

Thank you for your comments and suggestions. We have responded to them each as listed below. We also wish to point out that this first manuscript was submitted along with a second companion study ( Mudryk et al. also currently under review, https://doi.org/10.5194/egusphere-2023-3014). The focus of this first manuscript is to evaluate the extent to which the snow course and gamma reference data can be combined in order to increase total coverage across the North American continent. The focus of the second study is to use the combined reference information along with snow course information over Eurasia to provide a more pan-NH assessment of gridded product performance. This information may help clarify some of the choices we made with regards to aggregation distances as described below. More explicit reference linking these two manuscripts will be added to the introduction.

**It is easy for the snow water equivalent in mountainous areas to exceed 1000 mm. However, the 1000 mm SWE was excluded from the validation in the manuscript, did the authors calculate the amount of data for these exclusions, and did they affect the accuracy assessment of SWE in mountainous areas, where snow depth tends to be very large. I suggest the authors add a discussion of the relevant chapters in Uncertainty.**

While mountain SWE can indeed exceed 1000mm, observations above this limit represent only a very small proportion of the available reference data - 4% of mountain snow course and <2% of the total snow course observations. It does not apply to the airborne gamma data since its detection limit is ~1000mm. We added the proportion of data removed by applying this threshold.

**L124-125:** *"This threshold removed <2% of the snow course data (4% of the mountain data) and <0.2% of product SWE with coincident snow course observations."*

**As the UA SWE with the highest accuracy, its scatter plot does not show obvious scatter aggregation in the low-value area. So I would like to know if the number of verification points in Figure 1 is the same for each type of SWE product, please give the total number of verification points.**

The total number of data pairs after aggregation have been added to Figure 2 and the product names have been revised for consistency with Table 1.

The number of data pairs is smaller for UA SWE and Snow CCI because they do not cover the full spatial domain, and for JAXA AMSR2 and GLDAS 2.2 which are limited temporally.

[Figure]

*Figure 2: Product vs reference SWE density scatter for measurements > 0 and ≤ 1000 mm during February – April. See Table 1 for product names, descriptions, and time periods. Note that Snow CCI excludes areas of complex terrain, U Arizona is limited to CONUS, and JAXA-AMSR2 (2014-) and GLDASv2.2 (2003-) are limited temporally.*

**Line 125, How do authors retain two-thirds of these sites, and what are the retained principles?**

Text revised for clarity.

**L126-128:** *"We retained reference sites that have SWE estimates from two-thirds of the products listed in Table 1; this number is roughly equivalent to the number of products covering the full spatial and temporal domain."*

**Line 133, the authors highlighted the importance of preventing oversampling in spatially dense areas by limiting the sampling of snow course and gamma SWE to 100 km. However, considering that the resampled grid surpasses the dimensions of certain SWE datasets, could this potentially introduce additional sampling errors that might impact the validation results?**

Thank you for your comment. Both reviewers noted the lack of clarity in our description of the spatial aggregation method. We have reworked the description in the revised manuscript (~L134-146) to better describe our approach (see revised text in red further below). We also provide additional clarification immediately below on the rationale for aggregating and on the sensitivity of product statistics to the spatial aggregation distance.

To clarify your question regarding resampling, we did not 'resample' the data as our original text implied. Instead, we averaged the in situ data at the resolution of each product (obtained paired reference-product SWE values) and then aggregated product-reference pairs within a given search radius (100km radius, equivalent to a 200km search window). The spatial aggregation applied in our study is primarily an attempt to minimize biases in the resulting statistics due to the uneven spatial distribution of the reference data. As we show in the figure below (S1), while the choice of aggregation distance impacts the value of the statistics somewhat, it has little-to-no impact on product rankings, and increasing the aggregation distance generally improves product performance up to 200km or so (the increasing amount of aggregation makes the spatial scale of the reference data more consistent with that of most of the gridded products). Our choice of 100-200km for the aggregation scale was intended to obtain a relatively even spatial distribution of the reference data over North America. In addition, it keeps the spatial density of reference data over North American roughly proportional to that over Eurasia, a characteristic that was useful for our companion study noted at the beginning of these responses.

Figure **S1**, presents the validation metrics for spatial aggregation windows between 4km and 500km. The smaller windows (4,10,20,50km) loosely correspond to the resolutions of products tested and allows for better comparison between the aggregated data and native product grid. In general, product ranking remains similar across the range of spatial aggregation distances tested (the horizontal lines rarely cross each other); inter-product differences are minimally impacted by the aggregation. With some exceptions, product metrics improve with spatial aggregation distance as data are smoothed, although the redistribution of data is likely to also have an impact (i.e. less weight given to regions with spatially dense reference data). Unsurprisingly, the change in product metrics with spatial aggregation distance tends to be larger in mountain areas where SWE varies over shorter distances.

[Figure]

**Figure S1:** *Product metrics calculated for various aggregation windows (see Sections 3.1 and S0). Crosses show the product metrics calculated at each products' native grid (i.e. all in situ observations on a given date within a product grid cell are averaged together); the circles to the left of zero show the product metrics calculated for all reference-product pairs (no averaging or aggregation). The grey vertical shading at 200km highlights the metrics presented in the manuscript.*

We also propose to add the following to the main body of the text to better explain the purpose of the spatial aggregation step and the general application:

**L126-146:** *"Reference SWE was matched up in space and time with gridded SWE at the native product resolution. To reduce errors from mismatched water and ice masks, we retained reference sites that have SWE estimates from two-thirds of the products listed in Table 1; this number is roughly equivalent to the number of products covering the full spatial and temporal domain less one to allow for minor differences in product masks. For gamma SWE, we used the midpoint of each flight line for geolocation, which differs slightly from Cho et al. (2019; 2020) and Tuttle et al. (2018) who weighted the average of the gamma SWE footprint (using a fixed diameter of 330 meters assigned to each flight line) contained within each product grid cell. We found that both methods produced similar results, so we used the flight line midpoint for simplicity.*

*The reference data were averaged to the resolution of each product. Next, to reduce oversampling of areas with spatially dense networks, all product-reference pairs within sequential 200km windows were averaged (see Supplement Sect. S0). This averaging window corresponds to the range of non-mountain SWE variability (~150-250 km, Pulliainen et al. 2020). Snow course and gamma SWE were considered separately, and mountain measurements were separated from non-mountain. This aggregation approach aims to provide a more even distribution of product errors across landcover types and snow classes. Sensitivity analysis of various spatial aggregation windows between 4 km and 500 km showed little impact of window size on product ranking (Figure S1, limited to 300 km for display purposes). In general, product metrics improve with aggregation window size up to ~100 km but inter-product differences remain fairly consistent. We selected a 200 km aggregation window, as a compromise between sample size and spatial distribution. This approach, which effectively averages the reference data at the scale of the native product grid and then averages product errors within a larger area, is sufficiently flexible to enable the tests of covariates applied in Sections 4.3 through 4.5."*

*Pulliainen, J., Luojus K., Derksen C., Mudryk L., Lemmetyinen J., Salminen M., Ikonen J., Takala M., Cohen J., Smolander T., Norberg J. 2020: Patterns and trends of Northern Hemisphere snow mass from 1980 to 2018, Nature, 581, https://doi.org/10.1038/s41586-020-2258-0, 2020.*

We will also add supplemental text to detail the precise approach. We found that adding these details to the main text induced confusion for the reader.

*"As outlined in Section 3.1, we aggregated the reference data at the scale of the native product grid and then averaged the reference-product pairs within a larger window. Because the product grids do not overlay perfectly we did the following:*

*Sites within 100km of a base site were identified. If, within a given pool of matched reference sites, there were multiple reference-product data pairs within the same native product grid, these pairs were averaged. The mean product and reference SWE within each pool of data were then calculated. This process was repeated sequentially, starting with site ALE-05AA805 and ALE-05FA802 for snow course mountain and non-mountain respectively and AK101 and AB101 for gamma mountain and non-mountain respectively. Sites included in a search pool were dropped from the list and the window moved to the next site on the list. Snow course and gamma SWE were considered separately, and mountain measurements were separated from non-mountain."*

**Is Figure 2 a scatter plot obtained by sampling the snow course and gamma SWE to 100 km? Why not choose a smaller scale? Will this affect the accuracy of verification results?**

Yes, Figure 2 was produced using the aggregated data. As described above aggregating to 200km (100km search radius) minimally impacts the validation. Please also note that during revision we realized what we were describing as a 100km aggregation distance was actually the value used for the reference-product pair search radius and thus equates to a 200km search window. This error has been corrected.

**The abbreviations of ESn and CE5 in Figure 2 do not correspond to the abbreviations in Table 1. Please review the product abbreviations throughout the document and make sure they are aligned.**

Thank you for noticing these inconsistencies. We will change the abbreviations in Figures 2 and 8 to match those in Table 1 and will modify the abbreviation for ERA5-Snow in the table to ESn to avoid confusion with ERA5 (E5).

**Line 133-135: "Sensitivity analysis of various spatial aggregation distances between 50 and 200 km showed little impact of aggregation distance. We selected 100 km as a compromise between sample size and spatial distribution". How can we see "little impact of aggregation distance"? Will the sensitivity analysis results be considered for addition, and further explanation is needed for the selection basis of 50-200 km aggregation distance.**

As evidence of this claim, we now include the figure shown above in the supplementary material and refer readers to it.

**Since the North American reference SWE dataset was finally constructed in the article, please consider whether further additions are needed using this dataset to verify the SWE grid products mentioned in the article.**

A reference dataset very similar to what was used in our analysis is provided here: https://zenodo.org/records/10287093. There are additional data used in our analysis from the Ministère de l'evnrionment, la lute contres les changements climatiques, et de la Faune et des Parcs de Québec that we are unable to share publicly. These data were published following our initial manuscript submission. We have added reference to this dataset in the manuscript where appropriate.

We have reworked the description in the revised manuscript (~L134-146 and Supplement) to better describe our approach (see response and revised text to previous comment).

---

## Author Comment (AC3)

**Response to Reviewer #2 (Simon Gascoin)**

**This study presents an evaluation of several SWE products with spatial resolutions ranging from 4 km to over 100 km. These SWE products are constantly evolving and it is very useful to have an up-to-date assessment of their strengths and weaknesses, in particular to assess the impact of climate change on global snow mass. The reference data are airborne gamma and snow courses which represents a novelty compared to previous studies (L60 "a unified assessment of gridded SWE products using both reference datasets is lacking"). The analyzes are the result of significant work since 14 products were evaluated over a vast area with a large data set. This work is therefore of notable interest. In my opinion, Figure 2 alone is useful enough for this work to be published.**

We would like to thank Simon Gascoin for their feedback. We have responded to them each as listed below. This manuscript accompanies that of Mudryk et al. also currently under review, https://doi.org/10.5194/egusphere-2023-3014. We considered formally linking the two manuscripts for coordinated review at the journal level but ultimately we decided against this. We apologize for the lack of information concerning the second manuscript which may have made our objectives in this study clearer. More explicit reference linking these two manuscripts will be added to the introduction.

The objective of our manuscript, as stated on L66, is to assess the sensitivity of product performance to the choice of reference dataset. *"We investigate the agreement in reference SWE reported by the two reference datasets at various spatial and temporal scales and explore how the choice of reference dataset affects the accuracy assessment and overall performance ranking of the products."*

The analysis contained in the current manuscript was important to help understand how and where the choice of reference dataset influences the calculated product statistics. Our evaluation led us to conclude that, at least in non-mountain areas, we can use snow course and airborne gamma SWE estimates in concert. This conclusion allowed us to create a combined reference dataset that is then used to critically assess 23 gridded SWE products in Mudryk et al. (under review). The complementary coverage from the two types of reference data considered in our analysis was a key motivation for the analysis contained in our manuscript. Importantly, we are trying to design an evaluation scheme that is appropriate and relevant for the scale of the products being evaluated. That is, continental to hemispheric-scale SWE status and trends.

**However, the rest of the study is much less convincing in my opinion. The authors decided to analyze the impact of the reference dataset on the evaluation results. This gives e.g. scatterplots of correlation coefficients with legends indicating correlations of correlations (Fig. 5), difficult to understand and above all of an interest which escapes me. What do we learn about the SWE products from this?**

As noted above, the objective of this paper was not to learn about the SWE products from this analysis (again, this is covered in depth in the Mudryk et al. under review) but rather to understand the impact of the choice of reference dataset on product evaluation. We understand that the absence

of information about the companion manuscript may have created the false impression that the present manuscript's focus was a detailed analysis of the 14 SWE products listed in Table 1.

**Next, the authors conclude their study by presenting a combined benchmark dataset that is generated by an aggregation method that I did not understand\*.**

**The end of the reading leaves me with the impression that the authors used products to evaluate the validity of the observations which will ultimately be used to evaluate these products (cf. Sect 5.1, 5.3). Maybe I didn't understand correctly, but isn't there a form of circular reasoning here?**

The preliminary product evaluation was simply to screen out the worst performers to make sure their poor performance didn't unduly influence the subsequent analysis We then used the refined selection of products to assess the agreement in product statistics between the two types of reference measurements. The reference datasets have some amount of spatial overlap, but their coverage is mostly complementary. The aim was to determine how well the reference SWE and the product metrics (absolute values and product rankings) agree to help identify the conditions under which it would be reasonable to combine the two types of reference measurements as part of a single analysis of targeting continental or hemispheric-scale SWE.

**Another question: a product can be evaluated by observations of different types (with uncertainties and specific spatial characteristics), why aggregate these data in a multi-source composite? By aggregating the risk is to lose knowledge of the error associated with each observation.**

Although it is interesting to provide detailed analysis with individual reference data and at various spatial and temporal resolutions, it is also necessary to devise ways of summarizing such information for evaluations at larger spatial scales. Our combined dataset is one step towards achieving this goal. Detailed analysis of specific regions or reference measurements as you point out are important; our work is not intended to serve those purposes.

**In situ measurements of SWE in mountain areas can vary drastically on the scale of a few kilometers. The problem does not come from the observations but from the evaluated products which give a representation of the snow cover on a smoothed landscape. Taking into account the spatial variability of mountain SWE which is documented in numerous studies (a bibliographic analysis on this subject would have been useful), the date-to-date comparison of a SWE value in a region of 50 km x 50 km with a SWE value obtained by snow course seems very random. In fact, a snow course value taken in a region of 50 km x 50 km can be seen as a random draw from a SWE distribution which would likely extend from 0 to >500 mm w.e. The representativeness of this measurement can be assessed using the SWE semi-variogram. If the range of the SVG of the SWE is close to the resolution of the model then the comparison is well founded. Otherwise, one way to overcome these known biases could be to select observations which have altitudes close to the altitude of the model grid. Or to consider the anomaly of the SWE in relation to an interannual average SWE in order to remove the first order effect of the topography. Another option would be to consider the higher resolution Arizona dataset as the reference (after independently**

**evaluating it using in situ data, unless this has already been done), thereby aggregating that reference on the grid of each hemispherical products to facilitate their evaluation, including stratifying the residuals by elevation, land cover, etc.**

Thank you for your interest in mountain SWE evaluation. Part of the issue here may be due to the misunderstanding surrounding the aim of our paper and that of the companion manuscript, outlined above. We have tried to address your suggestions below.

In our original (and revised) manuscript we present that our validation approach may be the source of some of the issues in mountain regions:

> **L396-399:** *"As outlined in Sect. 3.2.4, in mountain and complex terrain, the relationship between SWE and elevation can result in large SWE gradients over short distances (i.e. less than a single product grid cell). In these regions, systematic differences in elevation between reference measurements and the centroid of a product grid could, therefore, produce validation errors that are a result the validation approach rather than the products themselves."*

As you mention, one approach to address the issue of representativeness of mountain SWE would be to threshold the data to those with similar elevations. We investigated this issue indirectly in Section 4.5 where we looked at the relationship between the agreement in product metrics and the magnitude of elevation biases. What we found was that the differences in SWE magnitude sampled by the two reference datasets override those of elevation bias (this was true when we reversed the order in which these thresholds are applied).

We chose to present analysis using the full reference dataset first because it allows us to then comment on the impact of elevation biases. As you rightly point out, when such biases are not accounted for, they can result in apparent product errors. We did consider thresholding the product-reference data pairs by elevation bias at the outset. This approach limited the available data pairs for comparison between snow course and airborne gamma reference datasets. In the end we felt that including all reference data provided an opportunity to first comment on the on the imperfections of a traditional validation approach, including those related to elevation differences.

**The representativeness of this measurement can be assessed using the SWE semi-variogram. If the range of the SVG of the SWE is close to the resolution of the model then the comparison is well founded.**

At your suggestion we have done some additional investigation of the representativeness of the in situ SWE via semi-variograms. As input, we used the mean March SWE at each snow course site, calculated over a 30year period (1990-2019).

A test region in Eastern North America [40°N-55°N, 65°W-90°W] (Fig. R2a) suggests a range of around 125-150km which is consistent with previous analysis of snow course data (e.g. Pulliainen et al. 2020). Results for the western mountain region (Fig. R2b) are less clear. The spatial distribution of the available snow course data is probably insufficient to resolve the true scales of mountain SWE variability. Our analysis seems to indicate that information at scales below ~2km

is not captured by the reference data and, as you rightly point out, there are important snow processes operating below that scale. Nonetheless, the mountain semivariograms suggests a range of <5km. This is smaller than the grid spacing of almost all products evaluated which suggests that the gridded products are too coarse to capture the smallest scale information provided by the reference data in mountain regions. However, by aggregating the reference data to larger scales we still expect a certain level of agreement between the reference data and the gridded products (see Figure S1). Importantly, our analysis is able to capture inter-product differences in mountain regions.

[Figure]

**Fig. R2a** Semi-variogram of mean March SWE from snow courses for the period 1990-2019 for East region [40°N-55°N, 65°W-95°W] suggests two scales of variability: one around ~150km and another around 450km. Lag is in metres.

[Figure]

**Fig. R2b** Semi-variogram of mean March SWE from snow courses for the period 1990-2019 for mountain regions west of 103°W excluding Alaska for max lag (a) 100,000m (100km) and (b) 500,000m (500km). Lag is in metres. Suggests small-scale range of <5km (a) and a second larger scale range around ~200km (b).

This finding will be clarified in the revised manuscript (new text in red). While we hope the semi-variograms shown below address your comment, we feel the addition of this material is not required in the revised manuscript.

> **Discussion 5.2 L442-450**: *"Aside from this product, a consistent high-level message from our analysis is that products perform considerably worse in mountain compared to non-mountain areas. The grid spacing of nearly all products evaluated is larger the mountain SWE autocorrelation length determined from snow courses (<~5km) (not shown) which suggests that the current suite of global reanalysis and EO products are too coarse to capture the smallest scale information provided by the reference data in mountain regions. The challenge of accurate SWE estimation from coarse-resolution gridded SWE products is a well-documented issue (Fang et al., 2022 and references therein; Kim et al., 2021; Liu et al., 2022; Snauffer et al., 2016; Terzago et al., 2017; Wrzesien et al., 2019). However, our analysis also shows that the choice of reference data may also contribute to poorer product performances, as demonstrated by the large discrepancy in product metrics computed with the two reference datasets in coincident mountain areas (Fig. 6). Importantly, despite these limitations, our analysis is able to capture inter-product differences in mountain regions."*

**Otherwise, one way to overcome these known biases could be to select observations which have altitudes close to the altitude of the model grid.**

- As outlined above, the impact of elevation differences were examined in Section 4.4 (now section 4.5).

**Another option would be to consider the higher resolution Arizona dataset as the reference (after independently evaluating it using in situ data, unless this has already been done), thereby aggregating that reference on the grid of each hemispherical products to facilitate their evaluation, including stratifying the residuals by elevation, land cover, etc.**

- Yes, we did try using the higher resolution UArizona dataset as a reference. A consideration for us was that the product only covers CONUS and we our goal was to establish a reference dataset that could be used in a larger analysis of the full Northern Hemisphere (Mudryk et al. under review). The UA dataset did not serve that purpose.

**...the authors conclude their study by presenting a combined benchmark dataset that is generated by an aggregation method that I did not understand\*.**

**\* This product is formed by a method that I do not understand : L132 "To avoid oversampling specific grid cells, we first aggregated reference sites within the same product grid cell (at the native resolution of the product grid) before aggregating to the 100 km spacing." See also Section 3.3.2: I have read this part several times and am unable to understand what is being done. It would have been useful to share the source code of the**

**analyzes (what does aggregation, resampling mean? average, median, bilinear interpolation? how is the centroid of the new data defined?)**

Both reviewers noted the lack of clarity in our description of the spatial aggregation method. We have reworked the description in the revised manuscript (~L134-146) and provided additional details in the Supplement to better describe our approach (see revised text in red further below). We also provide additional clarification immediately below on the rationale for aggregating and on the sensitivity of product statistics to the spatial aggregation distance.

The goal of our approach was to create a more even sample distribution across landcover types and snow classes. We did this by first averaging the in situ data at the resolution of each product (thereby obtaining paired reference-product SWE values) and then aggregating product-reference pairs within a given search radius (100km, equivalent to a 200km aggregation window).

The first step limits the weight given to specific grid cells having multiple coincident observations on the same date compared to those with only one observation. The second step limits sampling differences related to gridded product resolution (otherwise products with smaller grid spacing would have proportionally more reference-product data pairs in areas with a high density of reference observations compared to products with a coarser spatial resolution). As we show in the figure below, while the choice of aggregation window size impacts the value of the statistics somewhat, it has little-to-no impact on product rankings, and increasing the aggregation window size generally improves product performance up to 100-200km or so (the increasing amount of aggregation makes the spatial scale of the reference data more consistent with that of most of the gridded products). Our choice of 200km for the aggregation window was intended to obtain a relatively even spatial distribution of the reference data over North America. In addition, it keeps the spatial density of reference data over North American roughly proportional to that over Eurasia, a characteristic that was useful for our companion study noted at the beginning of these responses.

[Figure]

***Figure S1:*** *Product metrics calculated for various aggregation windows (see Sections 3.1 and S0). Crosses show the product metrics calculated at each products' native grid (i.e. all in situ observations on a given date within a product grid cell are averaged together); the circles to the left of zero show the product metrics calculated for all reference-product pairs (no averaging or aggregation). The grey vertical shading at 200km highlights the metrics presented in the manuscript.*

We also propose to add the following to the main body of the text to better explain the purpose of the spatial aggregation step and the general application (new text in red):

**L126-146:** *"Reference SWE was matched up in space and time with gridded SWE at the native product resolution. To reduce errors from mismatched water and ice masks, we retained reference sites that have SWE estimates from two-thirds of the products listed in Table 1; this number is roughly equivalent to the number of products covering the full spatial and temporal domain less one to allow for minor differences in product masks. For gamma SWE, we used the midpoint of each flight line for geolocation, which differs slightly from Cho et al. (2019; 2020) and Tuttle et al. (2018) who weighted the average of the gamma SWE footprint (using a fixed diameter of 330 meters assigned to each flight line) contained within each product grid cell. We found that both methods produced similar results, so we used the flight line midpoint for simplicity.*

*The reference data were averaged to the resolution of each product. Next, to reduce oversampling of areas with spatially dense networks, all product-reference pairs within sequential 200km windows were averaged (see Supplement Sect. S0). This averaging window corresponds to the range of non-mountain SWE variability (~150-250 km, Pulliainen et al. 2020). Snow course and gamma SWE were considered separately, and mountain measurements were separated from non-mountain. This aggregation approach aims to provide a more even distribution of product errors across landcover types and snow classes. Sensitivity analysis of various spatial aggregation windows between 4 km and 500 km showed little impact of window size on product ranking (Figure S1, limited to 300 km for display purposes). In general, product metrics improve with aggregation window size up to ~100 km but inter-product differences remain fairly consistent. We selected a 200 km aggregation window, as a compromise between sample size and spatial distribution. This approach, which effectively averages the reference data at the scale of the native product grid and then averages product errors within a larger area, is sufficiently flexible to enable the tests of covariates applied in Sections 4.3 through 4.5."*

*Pulliainen, J., Luojus K., Derksen C., Mudryk L., Lemmetyinen J., Salminen M., Ikonen J., Takala M., Cohen J., Smolander T., Norberg J. 2020: Patterns and trends of Northern Hemisphere snow mass from 1980 to 2018, Nature, 581, https://doi.org/10.1038/s41586-020-2258-0, 2020.*

We will also add supplemental text to detail the precise approach, along with Figure S1. We found that adding these details to the main text induced confusion for the reader.

*"As outlined in Section 3.1, we aggregated the reference data at the scale of the native product grid and then averaged the reference-product pairs within a larger window. Because the product grids do not overlay perfectly we did the following:*

*Sites within 100km of a base site were identified. If, within a given pool of matched reference sites, there were multiple reference-product data pairs within the same native product grid, these pairs were averaged. The mean product and reference SWE within each pool of data were then calculated. This process was repeated sequentially, starting with site ALE-05AA805 and ALE-05FA802 for snow course mountain and non-mountain respectively and AK101 and AB101 for gamma mountain and non-mountain respectively. Sites included in a search pool were dropped*

*from the list and the window moved to the next site on the list. Snow course and gamma SWE were considered separately, and mountain measurements were separated from non-mountain.*"

**In conclusion I think that the authors should rework their article in order to clarify their scientific objective but I am convinced that the analyzes already carried out have great value for the scientific community which studies snow mass on a global scale.**

Thank you for your constructive feedback. We hope that our responses have clarified the objective of our manuscript, and made clear how our analysis focused on the characteristics of the reference datasets complements the comprehensive assessment of 23 Northern Hemisphere SWE products in Mudryk et al. (under review). We have added text to clarify the study objectives, and added detail to our description of spatial aggregation which was a source of confusion for both reviewers.

**L68-70**: *This analysis assesses the feasibility of developing a combined (snow course + airborne gamma) continental-scale reference dataset, both for benchmarking the performance of gridded SWE products (see Mudryk et al. under review) and other hydroclimate applications.*

**L234-235:** *"A detailed analysis of these and nine other gridded SWE products over the Northern Hemisphere is provided in Mudryk et al. (under review)."*

**Minor comments**

**- Fig 7. I don't understand why the "full domain" histogram has lower values than the "restricted" histogram (e.g. in the 100-150 mm bin)**

The original Figure 7 showed the PDF and had different y-axis maximums. For ease of interpretation we will replace the PDF with total counts per bin as shown below.

[Figure]

*Figure 7: Reference SWE (top two rows) and elevation (bottom row) distribution for spatially and temporally restricted subset (top row) and the full domain (bottom rows) for mountain (left) and non-mountain (right). The spatial and temporal subset (top row) is the same reference data used to calculate the product statistics shown in Fig. 6 (hollow dots). Y-axis values are total counts.*

**- L80: Coterminus**

Coterminous

**- L174: "In mountain regions, large changes in elevation over short distances are common. (..) SWE decreases due to wind redistribution" A more in-depth bibliographic analysis on this subject in the introduction would be useful. By definition, "redistribution" does not reduce the SWE in average but increases its spatial variability. Think about precipitation gradients, blowing snow sublimation, avalanches, etc.**

We have revised the text to be more inclusive of processes contributing to mountain SWE variability and expanded our bibliographic analysis.

**L147-148:** *"Due to the well documented challenges in estimating and validating mountain SWE at coarse resolutions (Dozier et al. 2016; López-Moreno 2013; Wrzesien et al., 2019),..."*

**L188-193:** "*Mountain snowpacks exhibit considerable spatial and temporal) variability at short scales, associated with a suite of complex and interrelated factors including orientation, wind exposure, vegetation cover, slope, and elevation (e.g., Clark et al., 2011; Lopez-Moreno and Stähli 2008; Mott et al., 2010; 2018; Pomeroy et al., 1998; 2007 and references therein; Vionnet et al., 2021). Previous studies have often identified a positive correlation between elevation and SWE that tapers off at high elevations often above the treeline (e.g. Durand et al., 2009; Grünewald et al., 2014; Kirchner et al., 2014, Lehning et al., 2011; Rohrer et al., 1994;), which is above the elevation of most of our reference data.*"

*References*

*Clark, M. P., Hendrikx, J., Slater, A. G., Kavetski, D., Anderson, B., Cullen, N. J., Kerr T., Hreinsson E. O., and Woods, R. A.: Representing spatial variability of snow water equivalent in hydrologic and land-surface models: A review, Water Resour. Res., 47, W07539, https://doi.org/10.1029/2011WR010745, 2011.*

*Dozier, J., Bair, E.H., and Davis, R.: Estimating the spatial distribution of snow water equivalent in the world's mountains, WIREs Water, 3:461-474, https://doi.org/10.1002/wat2.1140, 2016.*

*Durand, Y., Giraud, G., Laternser, M., Etchevers, P., Mèrindol, L., and Lesaffre, B.: Reanalysis of 47 Years of Climate in the French Alps (1958-2005): Climatology and Trends for Snow Cover, J. Appl. Meteorol. Clim., 48, 2487–2512, https://10.1175/2009jamc1810.1, 2009.*

*Lehning, M., Gruenewald, T., and Schirmer, M.: Mountain snow distribution governed by an altitudinal gradient and terrain roughness, Geophys. Res. Lett., 38, https://10.1029/2011GL048927, 2011.*

*Lopez-Moreno, J.I. and Stähli, M.: Statistical analysis of the snow cover variability in a subalpine watershed: Assessing the role of topography and forest interactions, Journal of Hydrology, 348(3-4): 379-394, https://doi.org/10.1016/j.jhydrol.2007.10.018, 2008.*

*López-Moreno, J. I., Fassnacht, S. R., Heath, J. T., Musselman, K. N., Revuelto, J., Latron, J., Morán-Tejeda, E., and Jonas, T.: Small scale spatial variability of snow density and depth over complex alpine terrain: Implications for estimating snow water equivalent, Advances in water resources, 55: 40-52, https://doi.org/10.1016/j.advwatres.2012.08.010, 2013.*

*Mott, R., Schirmer, M., Bavay, M., Grünewald, T., and Lehning, M.: Understanding snow-transport processes shaping the mountain snow-cover, The Cryosphere, 4, 545–559, https://doi.org/10.5194/tc-4-545-2010, 2010.*

*Mott, R., Vionnet, V., and Grünewald, T.: The seasonal snow cover dynamics: review on wind-driven coupling processes, Front. Earth Sci., 6, 197, https://doi.org/10.3389/feart.2018.00197, 2018.*

*Pomeroy, J. W., Gray, D. M., Shook, K. R., Toth, B., Essery, R. L. H., Pietroniro, A., and Hedstrom, N.: An evaluation of snow accumulation and ablation processes for land surface modelling, Hydrol. Process., 12, 2339–2367, [https://doi.org/10.1002/(SICI)1099-1085(199812)12:15<2339::AID-HYP800>3.0.CO;2-L](https://doi.org/10.1002/(SICI)1099-1085(199812)12:15<2339::AID-HYP800>3.0.CO;2-L), 1998.*

*Pomeroy, J. W., Gray, D. M., Brown, T., Hedstrom, N. R., Quinton, W. L., Granger, R. J., and Carey, S. K.: The cold regions hydrological model: a platform for basing process representation and model structure on physical evidence, Hydrol. Process., 21, 2650–2667, [https://doi.org/10.1002/hyp.6787](https://doi.org/10.1002/hyp.6787), 2007.*

*Rohrer, M., Braun, L., and Lang, H.: Long-Term Records of Snow Cover Water Equivalent in the Swiss Alps 1. Analysis, Nordic Hydrology, 25, 53–64, 1994.*

*Vionnet, V., Marsh, C.B., Menounos, B., Gascoin, S., Wayand, N. E., Shea, J., Mukherjee, K., and Pomeroy , J.W.: Multi-scale snowdrift-permitting modelling of mountain snowpack, The Cryosphere, 15, 743-769, [https://doi.org/10.5194/tc-15-743-2021](https://doi.org/10.5194/tc-15-743-2021), 2021.*

*Wrzesien, M.L., Pavelsky, T.M., Durand, M.T., Dozier, J., and Lundquist, J.D.: Characterizing biases in mountain snow accumulation from global data sets, Water Resour. Res., 55, 9873–9891, [https://doi.org/10.1029/2019WR025350](https://doi.org/10.1029/2019WR025350), 2019.*

**Fig. 2 legend: logarithmic or lognormal?**

Corrected to logarithmic. Thank you.

**Fig. 4 I would add the units to the RMSE and bias**

Units will be added to uRMSE and bias and rounded to nearest mm for space as shown below.

[Figure]

**a) Same date within various distances**

[Figure]

**b) Within 25 km and various temporal lags**

- What are the t-tests used for in this study? I missed it.

As stated in Methods 3.2 L157-159: *"For each of these covariates, a difference of means test (two-sided independent student t-test) was applied to determine whether the mean product metrics calculated using snow courses are different from those obtained with airborne gamma, using a significance level of 95%."*

- Fig. 8 is missing the x axis label

X-axis label will be added to the revised manuscript and to its partner Figure S4.

[Figure]

- L338: the bias decreases not increases (it is negative)

Revised to: Bias and uRMSE *magnitude* increase

- there are two sections 5.2

We apologize for the minor errors and inconsistencies. Section numbering and figures will be reviewed for consistency.

---

## Author Response (AR2)

Dear Simon Gascoin,

Thank you for your constructive feedback. We have revised the manuscript as outlined below.

I appreciate the responses by the authors, which helped me better understand the scope of this study. I hope it will help other readers too. I noted a few additional questions/suggestions below.

1) The semivariograms could be included in the manuscript (or in Supplement). This analysis can be used to strengthen this sentence in conclusion ("The strong agreement in non-mountain areas occurs because the reference SWE measured by gamma or snow courses agrees up to the scale of most gridded products evaluated")

We have added the semivariograms to the Supplement and referenced these figures in the main text.

2) I cannot agree with the implicit logic of this new sentence: "Importantly, despite these limitations, our analysis is able to capture inter-product differences in mountain regions". The fact that there are inter-product differences in the evaluation of the products is not a proof that the reference data were correct.

Sentence removed

3) In Supplement, I still struggle to understand the method of aggregation.
3.1) What is a "base site"?
3.2) This sentence confused me "Sites included in a search pool were dropped from the list and the window moved to the next site on the list." Does it mean that the final product is different if the first site in the list is different?
3.3) It would be useful to explicit what is meant by "within the same native product grid". Is it defined according to a radius or to the actual shape of the grid cell (I ask because both options were mentioned in the response letter "a given search radius (100km, equivalent to a 200km aggregation window)"). Some aggregation algorithms even consider points that are not within the target cell (e.g. inverse distance weighting, or kriging). Sorry to be picky here but it is a key algorithm of the final dataset (otherwise sharing the code would help).

**Revised supplement text:**

The following describes how we performed the data aggregation generalized in Section 3.1. As outlined in Section 3.1, we averaged all reference-product pairs within the same product grid at its native resolution and then averaged the reference-product pairs within a larger area. Because the product grids do not overlay perfectly we did the following:

> Starting with a single reference site all reference sites within 100 km of this site were identified. If, within a given pool of matched reference sites, there were multiple reference-product data pairs within the same native product grid, these pairs were averaged. The mean product and reference SWE within each pool of data was then calculated. This

process was repeated sequentially, starting with site ALE-05AA805 and ALE-05FA802 for snow course mountain and non-mountain, respectively, and AK101 and AB101 for gamma mountain and non-mountain, respectively. Sites included in a search pool were dropped from the list and the window moved to the next site on the list. Snow course and gamma SWE were considered separately, and mountain measurements were separated from non-mountain. Due to the nature of the aggregation method, the result will differ slightly if starting with site other than that listed above.

4) In the conclusion, it may be useful to clarify or rephrase this statement "targeted approaches are required to validate SWE products in mountain regions" since mountain SWE products were actually evaluated and ranked in the companion paper.

Text removed.